DOI: 10.1038/s41467-018-06940-5　　OPEN

# Ballistic tracks in graphene nanoribbons

Johannes Aprojanz[1], Stephen R. Power [2,3,4], Pantelis Bampoulis[5,6], Stephan Roche[2,7], Antti-Pekka Jauho[8], Harold J.W. Zandvliet[5], Alexei A. Zakharov[9] & Christoph Tegenkamp[1,6]

High quality graphene nanoribbons epitaxially grown on the sidewalls of silicon carbide (SiC) mesa structures stand as key building blocks for graphene-based nanoelectronics. Such ribbons display 1D single-channel ballistic transport at room temperature with exceptionally long mean free paths. Here, using spatially-resolved two-point probe (2PP) measurements, we selectively access and directly image a range of individual transport modes in sidewall ribbons. The signature of the independently contacted channels is a sequence of quantised conductance plateaus for different probe positions. These result from an interplay between edge magnetism and asymmetric terminations at opposite ribbon edges due to the underlying SiC structure morphology. Our findings demonstrate a precise control of transport through multiple, independent, ballistic tracks in graphene-based devices, opening intriguing pathways for quantum information device concepts.

[1] Institut für Physik, Technische Universität Chemnitz, 09126 Chemnitz, Germany. [2] Catalan Institute of Nanoscience and Nanotechnology (ICN2), CSIC and The Barcelona Institute of Science and Technology, Campus UAB, Bellaterra, 08193 Barcelona (Cerdanyola del Vallès), Spain. [3] Universitat Autònoma de Barcelona, 08193 Bellaterra (Cerdanyola del Vallès), Spain. [4] School of Physics, Trinity College Dublin, Dublin 2, Ireland. [5] Physics of Interfaces and Nanomaterials, MESA+Institute for Nanotechnology, University of Twente, 7522 NH Enschede, The Netherlands. [6] Institut für Festkörperphysik, Leibniz Universität Hannover, 30167 Hannover, Germany. [7] ICREA, Institució Catalana de Recerca i Estudis Avançats, 08070 Barcelona, Spain. [8] Center for Nanostructured Graphene (CNG), DTU Nanotech, Technical University of Denmark, DK-2800 Kongens Lyngby, Denmark. [9] MAX IV Laboratory and Lund University, 221 00 Lund, Sweden. Correspondence and requests for materials should be addressed to C.T. (email: christoph.tegenkamp@physik.tu-chemnitz.de)

Epitaxial graphene layers hold great potential for advanced interface engineering. The homogeneity of large graphene layers grown on SiC(0001) make them even suitable for quantum Hall metrology applications[1–4]. The selective growth of graphene by sublimation on the sidewalls of SiC mesa structures produces graphene nanoribbons (GNR) of excellent quality[5–12]. These ribbons have well-defined edge geometries—the realisation of which has presented an insurmountable challenge for many nanostructure fabrication and growth techniques, and which to date has only been partially achieved by chemical unzipping of nanotubes or self-assembly procedures[13–17]. The characteristic hallmark of sidewall ribbons lies in their μm-scale room-temperature ballistic transport with a single-channel conductance $e^2/h$ that is probe-spacing and temperature independent[1,3]. Interference effects in nanoconstrictions previously indicated the edge nature of the exceptional ballistic channel[18].

Extending these concepts further, we now report how the asymmetric edge morphology of sidewall ribbons gives rise to multi-channel ballistic wires. We achieve a direct visualization and characterization of multiple spatially-segregated ballistic modes on the nanometer scale. Furthermore, the explicit consideration of zigzag-edge magnetisation and transverse electric field effects within tight-binding calculations captures in detail the formation and localization of the experimentally-observed edge and bulk channels.

## Results

**Epitaxial zigzag GNR.** The geometry of a SiC facet with a GNR is depicted in Fig. 1a. Density functional theory (DFT) and transmission electron microscopy (TEM) have revealed that graphene growth is seeded at trenches close to the lower edge of the SiC facet structure, while the top of the ribbon merges into the buffer layer above the mesa[19–22]. For mesa structures running along the [$1\bar{1}00$]-direction and with trench depths of around 20 nm, SiC($11\bar{2}n$) facets approximately 40 nm wide with an inclination of 25–30° are formed during annealing. However, it should be noted that the SiC-sidewalls easily refacet, i.e. forming smaller faceted subareas, at temperatures where Si sublimation and graphene growth sets in refs.[7,23]. Recent optimization of growth conditions allows these energy-driven instabilities of the SiC($11\bar{2}n$) facets to be suppressed[24] (see Supplementary Figure 1). The scanning tunneling microscopy (STM) images in Fig. 1b, c show the SiC facet to be almost completely overgrown by graphene in zigzag orientation, with only the top part revealing signs of step-bunching effects (see also Supplementary Information, SI). In situ two-point probe (2pp) transport measurements were used to characterise the long-ranged quantum-transport properties in detail at room temperature in ultrahigh vacuum. The characteristic value of $R = h/e^2 \approx 26$ kΩ is measured when both probes are located on the ribbon with a separation of 2 μm, as shown in Fig. 1d and in full agreement with prior reports[1,3,18].

**Spatially resolved transport measurements.** To gain insight into the electronic structure variations across the ribbon width, we have performed spatially-resolved in situ transport experiments using a STM/scanning electron microscopy (SEM) system with two probes in ohmic contact. One tip was blunt and covered the entire ribbon width, whereas a second, sharper tip was moved transversely across the ribbon at a fixed probe-to-probe distance (Fig. 2a, b) as small as 70 nm. The correlated microscopy with SEM and STM enabled us to measure reliably the transport with ultra-small probe spacings (see Supplementary Figure 2). Figure 2c shows a conductance of $e^2/h$ when the mobile tip connects to the lower edge of the ribbon, corresponding to transport only through the exceptional edge channel. As the tip moves from edge

to the bulk, two higher conductance plateaus appear, whose values correspond closely to step sizes of $4e^2/h$ suggesting transport through additional four-fold degenerate ballistic channels. The corresponding IV-curves taken at these three distinct sites are given in the inset of Fig. 2c. The sequential appearance and disappearance of the additional channels is robust and reproducible, as demonstrated in Fig. 2d, which shows repeated sweeps with the mobile probe in both directions. The mean free path lengths of bulk channels in confined graphene nanostructures of this kind are of the order of 100 nm (ref.[25], see Supplementary Note 1). This prevented previous studies, with probe separations greater than 100 nm, from discerning the novel ballistic characteristics of higher order channels in sidewall ribbons. We note that the sharp tip still has a radius of the order of 40 nm, so that a transverse sweep of the tip across the ribbon from the bottom edge captures a cumulative effect as first a single edge channel, and then additional bulk channels, are contacted by the tip.

The sharp onset of the single-channel conductance with first a contact between tip and ribbon unambiguously demonstrates the location of the exceptional channel at the lower edge of the GNR, consistent with the previous characterisation of nanoconstrictions[18]. Its degeneracy and location are also consistent with a fully spin-polarised zigzag edge state[26–28]. The $4e^2/h$ conductance steps, on the other hand, are suggestive of transport through spin-degenerate and valley-degenerate confinement-induced sub-bands, such as those expected for pristine zigzag nanoribbons. The presence of two such steps indicates either contributions from two sub-bands, or that significant band-bending occurs to allow a single sub-band to cross the Fermi level multiple times. We further note that nanoribbon sub-bands are normally expected to be delocalised across the entire ribbon width, so that an increase of the contact area between the tip and ribbon should lead to a steady increase in conductance without significant step features. Quantised conductance plateaus are generally only expected when the electron density is varied to change the number of bands crossing the Fermi level. However, from Fig. 1b it is clear that the upper edge of the ribbon merges into a buffer layer structure present on the flat SiC(0001) parts. Significant charge transfer at this interface, analogous to the n-type doping of epitaxial graphene[29], can result in an inhomogeneous potential across the ribbon width, corresponding to a strong effective transverse electric field and thus leading to band-bending effects. A similar effect has also recently been observed at lateral WSe$_2$-MoS$_2$ heterojunctions[30]. We will demonstrate below that band-bending can account for both the segregation and degeneracy of the bulk transport channels observed in our measurements.

**Conductive-AFM measurements.** The spatial distribution of the various transport channels across the GNR, suggested by 2pp measurements, is further confirmed by conductive atomic force microscopy (c-AFM), which gives access to a direct real-space imaging of the transport channels. The AFM topography (Fig. 3a, b) once more uncovers a uniformly overgrown and smooth facet structure. Moreover, the simultaneously measured current image reveals multiple extended conductive channels parallel to the ribbon edge (Fig. 3a). A cross section (Fig. 3b) shows that a large local current flows at the lower edge with smaller currents across the rest of the ribbon. As evidenced by multiple IV-measurements, summarised by the histogram shown in Fig. 3d, the quantised conductance of the edge channel is once more reproduced (cf. Fig. 3c, d). The measurement of the characteristic quantum conductance value $e^2/h$ at large probe spacings under ambient conditions strongly underlines the robustness of the exceptional edge channel.

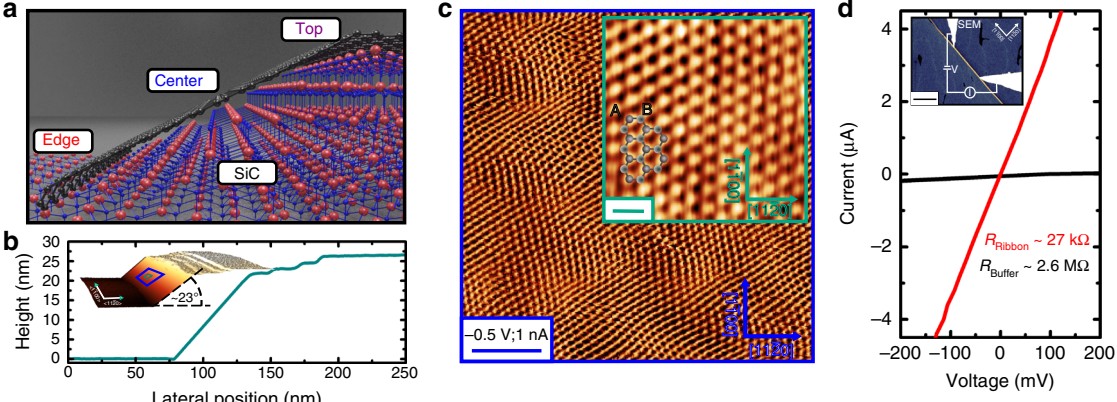

**Fig. 1** Ballistic transport in graphene sidewall ribbons on SiC mesa structures. Graphene nanoribbons (GNRs) can be selectively grown on SiC sidewalls as sketched in **a**. **b** Sequence of STM measurements performed at room temperature show the entirely overgrown SiC facet areas (+2 V, 0.5 nA, semi-insulating SiC). **c** High-resolution STM showing the overgrowth of the SiC facet and zigzag orientation (inset). The scale bars indicate a length of 2 nm (blue) and 0.4 nm (green). **d** The IV-curves measured in a two point probe assembly (2pp) clearly reveal a resistance of $h/e^2$ on the GNR for a probe distance of 2 μm. The GNR can be easily seen also in SEM (inset, doped-SiC, scale bar, 1 μm). By means of a 4-tip STM the ribbons are contacted for in situ transport measurements

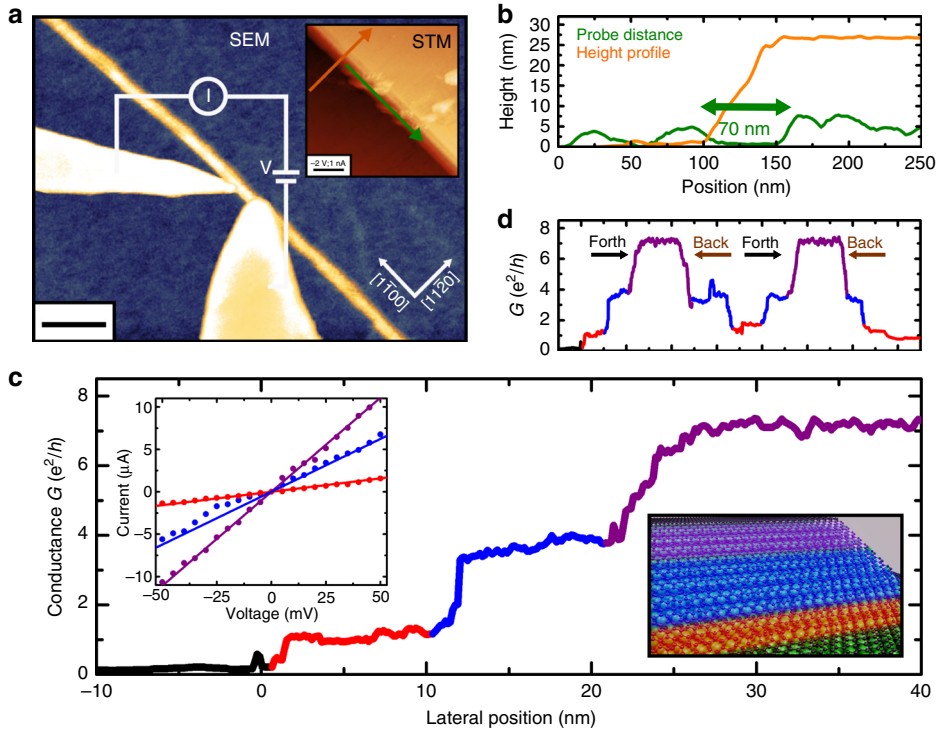

**Fig. 2** Spatially resolved 2pp transport measurements. **a** SEM image of a ballistic ribbon with an overlaid schematic showing a blunt and sharp tip with a scale bar of 300 nm. Inset: STM topography taken after transport measurements (scale bar, 100 nm). **b** Line scans across the ribbon, directions are indicated in the inset of **a**. **c** Conductance $G$ measured for a fixed distance $L = 70$ nm, while the sharp tip was moved across the ribbon starting from the lower edge ($U = 200$ mV). Inset: IV-curves measured at bottom, middle and top of the GNR. **d** The sequence of the channels can be reversibly measured by moving the ohmically contacted tip forward and backward

**Tight-binding calculations**. To analyse the exceptional transport features and understand the exact origin of the various modes, we have performed full-scale quantum-transport simulations of zigzag-edged nanoribbons. Excellent agreement with experimental measurements is obtained (Fig. 4a) when these calculations account for both edge magnetism and a spatial segregation of the bulk eigenmodes induced by asymmetries between the lower and upper edges of the ribbon, which connect to SiC(0001)

and the buffer layer, respectively. Previous studies support the formation of a spin-polarised state at a zigzag edge[27,31] and its robustness at the graphene/SiC(0001) interface[32]. In our model, we restrict the presence of edge magnetism to the lower edge of the ribbon, as strong doping effects and the lack of a sharp zigzag interface are expected to quench such behaviour at the top edge. To account, in a general way, for inhomogeneous potentials that arise due to the merging of the upper edge with the buffer region,

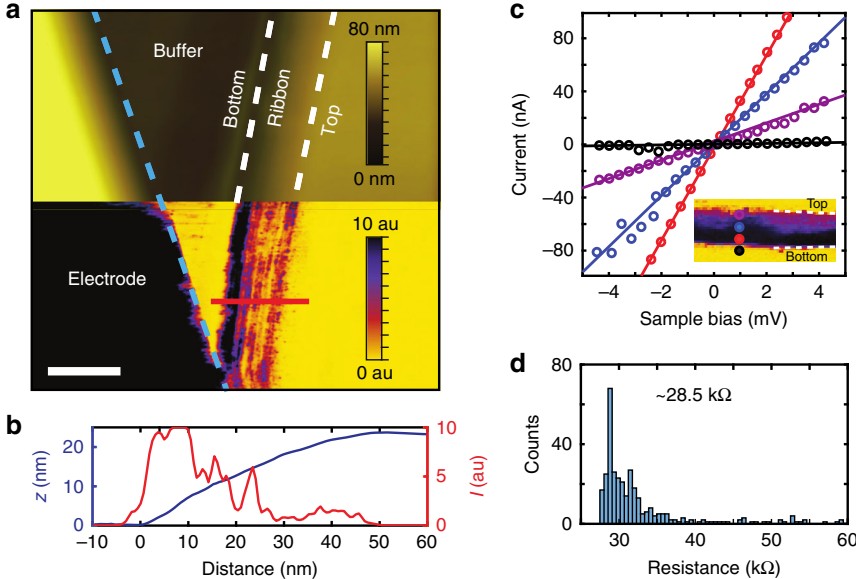

**Fig. 3** Direct imaging of the current channels in ballistic GNRs. **a** Top: Topographic AFM image of a GNR recorded with a conductive Pt tip. Bottom: the simultaneously recorded current image (sample bias 30 mV), demonstrating that the bottom of the ribbon is significantly more conductive than the top. The scale bar corresponds to a length of 50 nm. **b** Current and topography cross sections measured across the GNR indicated with the white line in **a**. **c** IV-curves recorded in contact mode at the locations indicated at the inset. **d** The histogram of the resistance values measured on the ribbon of the inset of **c**. The AFM measurements were performed under ambient conditions at 300 K

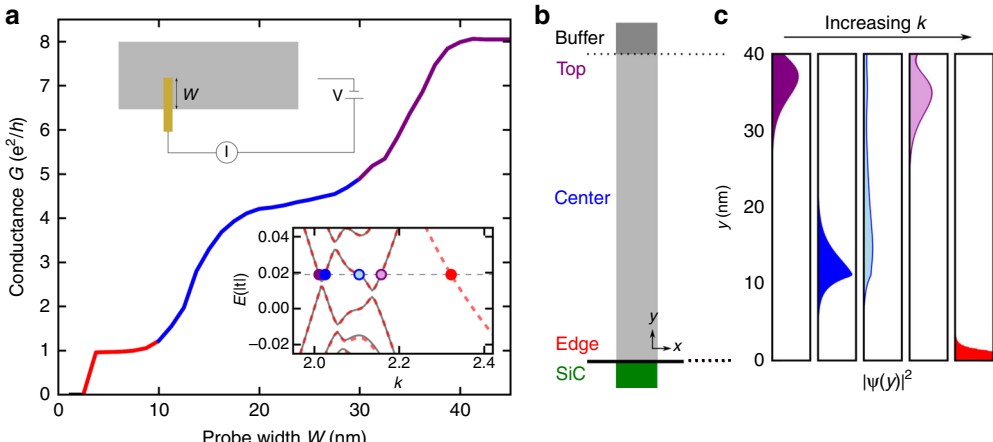

**Fig. 4** Tight-binding model of the edge and bulk channels. **a** Simulated two-point conductance as a function of the width of the mobile probe in contact with the ribbon, as shown schematically in the upper inset, capturing the characteristic stepwise features from experiment (see Fig. 2c). Lower inset: Band structure for spin-up (grey) and spin-down (red, dashed) electrons for a 188-ZGNR with edge magnetism and asymmetric potential terms. The dashed horizontal line shows the Fermi energy considered in the other panels, with the band crossings highlighted. **b** Schematic of the GNR. **c** Real-space projections of the states contributing to transmission (e.g. those at the corresponding crossing points in **a**) across the ribbon width. The blue and purple states are spin-degenerate, whereas the red state corresponds to the spin-polarised channel at the lower edge of the ribbon

and consequent charge transfer at this interface, we include a transverse electric field term which shifts the Fermi energy of the upper edge by approximately 0.5 eV relative to the lower edge. Monolayer graphene on SiC has been shown to be n-type doped by unsaturated bonds at the SiC interface[29,33]. The merging of the top edge of the ribbon into the buffer layer should result in a similar local doping scenario. In addition, ab initio studies[32] of narrow nanoribbons with both edges bonded to the Si-face of SiC (0001) reveal that hybridization quenches states other than the magnetic edge state, leading to secondary gap opening near the Fermi energy. Only the lower edge in our system has such a bond, so we reproduce the effect by adding a sublattice-dependent gap-opening term only at sites near this edge (see Methods).

The experimental three-plateau feature is accurately captured by our simulations once all the terms discussed above are included (Fig. 4a). Spin polarisation is required for an edge-localised state contributing $e^2/h$ to the conductance. The additional terms impose a spatial segregation of channels leading to step-like transitions as a function of probe position. The gap-opening term isolates the magnetic edge state from the bulk channels so that it can be resolved separately, whereas the effective electric field breaks the uniform distribution of bulk states across the ribbon width. For a small field, this term segregates valence and conduction band states towards opposite edges of the ribbon, but, for larger values, bands near the Fermi energy contain of an admixture of states with both conduction

and valence band characteristics. This leads to a distinctive W-shape bending of the low-energy bands, as evidenced by the band structure in the lower inset of Fig. 4a and further analysed in Supplementary Note 3. Within this energy region, current-carrying states from the same band can be localised at opposite edges of the ribbon (Fig. 4c), belying a mix of conduction and valence band characteristics. We note that the spatial segregation and degeneracies of the bulk experimental transport channels are entirely consistent with a single bent sub-band with spin and valley degeneracies. They are however not consistent with transport through multiple sub-bands since such a scenario tends to cluster states entirely along one edge. We note that a wide-range of gap-opening and transverse field parameters give rise to spatially-separated channels such as those reported here, supporting the robustness of these transport signatures (see Supplementary Figure 5). Furthermore this behaviour persists over a wide range of Fermi energies near the Dirac point (see Supplementary Figure 4).

## Discussion

In conclusion, we have demonstrated that the edge morphology-induced asymmetry between the upper and lower edges of side-wall nanoribbons generates a unique regime of segregated transport channels. Using an in situ multi-probe setup with significantly reduced probe separations, we have been able to sequentially contact individual channels within the ballistic quantum-transport regime. This has enabled selective transport measurements through various combinations of edge and bulk modes, and gives rise to an extraordinary series of quantised conductance plateaus as the probe position is varied. Our results highlight that edge morphology is crucial to fully understand mesoscopic transport and to further utilise such phenomena in device architectures. The availability of multiple, selectable quantum-transport channels opens intriguing possibilities for information transfer and logic applications, whilst the strong dependence of transport on the position of the mobile probe suggests methods of investigating strain or vibrational properties. Finally, our work reinforces the particular strength of two-point probe techniques in characterising systems with an interplay of edge and bulk transport phenomena. We expect that similar approaches can shed new light on a range of systems where such interplays occur, including the interfaces of lateral hetero-structures, systems with emergent topological effects, and the quantum Hall effect.

## Methods

**Preparation of GNRs**. For the growth of GNRs we use SiC wafers commercially purchased from SiCrystal AG (n-doped) and II-IV Deutschland (semi-insulating). The doped SiC substrates were flattened by using the face-to-face heating method and direct current heating, whereas the semi-insulating wafers were epi-ready[3,34]. Subsequently mesa structures with lateral dimensions between 1 and 8 μm and a height of around 20 nm were defined by using standard UV lithography and reactive ion etching (gas mixture 20/7 $SF_6/O_2$, power 30 W). GNRs were grown exclusively on the sidewall of the mesa following standard recipes[3,5]. The selective growth of GNRs was carried out both by heating in our face-to-face heater as well as by sophisticated RF induction furnaces[24].

**In situ transport measurements**. We used a nanoprobe system (Omicron) for all in situ transport and STM experiments. It is equipped with four individual STM tips and a high-resolution Gemini SEM, allowing a precise navigation of the tips for in situ transport measurements and gentle feedback controlled approach. After switching off the feedback, the tips were lowered to the sample surface (by 2 nm) while checking the contact resistance until stable contact is reached. Tip residuals on the ribbons are seen when lowering by 15 nm. This mode was used in order to deduce the correct probe distances. All transport experiments in this study were done in a two-point probe (2pp) configuration with electrochemically etched tungsten tips. Before characterization, the GNR-samples were degassed in situ at 870 K for several hours. For further details see, e.g. ref. [3].

**Conductive-AFM**. AFM imaging was done in contact mode with an Agilent 5100 AFM (Agilent) and a RHK AFM/STM (BeetleTM, RHK Technology) in $N_2$ environment by continuously purging the AFM environmental chamber with $N_2$ gas. For current imaging (c-AFM), we used conductive Pt tips (12Pt400B-10, Rocky Mountain Nanotechnology) with a nominal spring constant of 0.3 N/m and a resonance frequency of 4.5 kHz. In our setup the tip is grounded and a bias voltage is applied to the GNR. In order to complete the electrical connection, to investigate charge transport along the nanoribbons, and to minimize contributions from the underlying SiC substrate, a Cr(5 nm)-Pt(35 nm) film is deposited at one end of the GNR and acts as the second electrode. In addition, lateral force microscopy (LFM) images were recorded simultaneously with the topography and current images, by measuring the torsion of the cantilever during scanning. The positioning of the AFM cantilever was controlled by optical micrsoscopy. All c-AFM investigations were made on GNRs fabricated on semi-insulating 6H-SiC(0001). In addition, also AFM measurements using ultra-sharp diamond tips were performed (cf. Supplementary Note 2, Supplementary Figure 3).

**Tight-binding model**. The electronic properties of the ribbon structures were simulated using a nearest-neighbour tight-binding Hamiltonian of the form

$$H = \sum_{i,\sigma} \varepsilon_{i,\sigma} \, \hat{c}_{i\sigma}^\dagger \hat{c}_{i\sigma} + t \sum_{<ij>,\sigma} \hat{c}_{i\sigma}^\dagger \hat{c}_{j\sigma} \,, \tag{1}$$

where $i$, $j$ are atomic site indices and $\sigma$ is a spin index, $< ij >$ indicates a restriction of the sum to nearest-neighbour sites only and $t = -2.7$ eV is the nearest-neighbouring hopping parameter. The onsite parameter $\varepsilon_{i,\sigma}$ is a sum of three terms $\varepsilon_{i,\sigma} = \varepsilon_{i,\sigma}^M + \varepsilon_i^G + \varepsilon_i^F$, each of which are position dependent, and correspond to contributions from edge magnetism (M), gap-opening near the lower edge (G) and the electric field (F), respectively. $\varepsilon_{i,\sigma}^M = \mp \frac{Um_i}{2}$ is a spin-dependent potential arising from a self-consistent mean-field approximation of the Hubbard model for the local magnetic moments $m_i$, and the on-site Hubbard parameter $U = 1.33|t|$ chosen has previously given good agreement with ab initio calculations[35]. This parameter is set to zero in the upper part of the ribbon. $\varepsilon_i^G = \pm \frac{\Delta_M}{2}$ is a sublattice mass term applied to a region approximately 10 nm wide near the lower edge of the ribbon which suppresses bulk states in an energy window of $\Delta_M \sim 0.2|t|$ around the Fermi energy. This mimics the previously-noted effects of hybridization with the SiC (0001) surface[32]. $\varepsilon_i^F$ varies linearly from $-0.1|t|$ at the upper edge to $0.1|t|$ at the edge of the sublattice gap region, including the role of the effective transverse electric field across the ribbon due to doping effects from the buffer region at the upper ribbon edge.

The 2pp transmissions are given by the Caroli formula[36]

$$T_{ij} = \mathrm{Tr}\left[ G^R \Gamma_b \, G^A \Gamma_a \right] \,, \tag{2}$$

where $G^R$ and $G^A$ are the (recursively calculated) retarded and advanced Green's functions respectively of an infinite nanoribbon system, and $\Gamma_{a(b)}$ is the broadening matrix associated with lead $a(b)$[37]. The use of zero-bias linear response techniques is justified by the independence of the experimental conductance on the bias voltage magnitude, as evident from the inset in Fig. 2c. The larger probe is modelled as one of the semi-infinite extensions of the nanoribbon, whereas the finite size probe is included via an effective self-energy $\Sigma_{metal} = -i|t|$ added to the sites in a rectangular region of varying width and constant length 1 nm to which the metallic tip couples (see Fig. 4a).

## Data availability

Authors can confirm that all relevant data are included in the paper and/or its supplementary information files. The underlying data used to generate the figures and conclusions in the paper are available from the corresponding author on reasonable request.

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

## Acknowledgements

Financial support by the Deutsche Forschungsgemeinschaft (Te386/12-1 and Te 386/13-1 (FlagEra Tailspin project)) is gratefully acknowledged by J.A. and C.T. P.B. and H.J.W. Z. thank the Stichting voor Fundamenteel Onderzoek der Materie (FOM, FV157 14TWDO07) for financial support. We acknowledge N. Vinogradov and Thi Thuy Nhung Nguyen for STM experiments and J. Schommartz for technical support. S.R.P. acknowledges funding from the European Unions Horizon 2020 research and innovation programme under the Marie Skodowska-Curie grant agreement No 665919 and from the Irish Research Council under the laureate awards programme. S.R. acknowledges funding from the Spanish Ministry of Economy and Competitiveness and the European Regional Development Fund (project no. FIS2015-67767-P MINECO/FEDER, FIS2015-64886-C5-3-P) and the European Union Seventh Framework Programme under grant agreement no. 785219 (Graphene Flagship). ICN2 is funded by the CERCA Programme/Generalitat de Catalunya and supported by the Severo Ochoa programme (MINECO, Grant. No. SEV-2013-0295). Research at DTU is supported by the Danish National Research Foundation, Project No. DNRF103. A.Z. acknowledges the Swedish Research Council (Vetenskapsrådet) for the Tailspin project support.

## Author contributions

A.Z. and J.A. fabricated the samples and J.A., P.B. and A.Z. performed the measurements. C.T. conceived and designed the experiment. S.R.P. performed the calculations. J.A, P.B., A.Z. and C.T. analyzed the data. All authors discussed the results and commented on the manuscript.

## Additional information

**Competing interests:** The authors declare no competing interests.

