## [Peer Review File · Nature Communications]

Reviewers' comments:

Reviewer #1 (Remarks to the Author):

The observation of quantized ballistic transport at ambient conditions and over large distances in one-dimensional structures formed in epitaxial graphene at the sidewalls of SiC substrate is highly intriguing. This work makes an important advance in investigating this phenomenon by means of spatially resolved two-probe measurements performed in a highly sophisticated experimental setup. In particular, this paper shows precise localisation of different ballistic channels and their spatial segregation. In my opinion, the experimental part of this work merits publication in Nature Communications on its own. The interpretation of experiments is supported by a computational model that leaves more questions. Overall, the model shows agreement with experimental data, but I have a strong impression that any small change in model parameters may produce qualitatively different results. Furthermore, the choice of some of these parameters is hard to justify. I would encourage the authors to answer the questions below, provide arguments in support of the robustness of the results to small changes of input parameters, and possibly tone down the unambiguity of the theoretical interpretation of experiments. The questions are:

1. As Fig. 4a inset shows a portion of band structure with a precise location of the Fermi level. Is this choice of Fermi level somehow justified? How the results would change if the Fermi level is displaced by a certain amount (e.g. 0.05 t)? It would be beneficial to the readers to present also the band structure plotted in a broader energy range and in the entire range of momenta.
2. The calculations seem to be performed at zero-bias conditions. Does this correspond to the experimental conditions?
3. What is the physical mechanism behind the mass term?
4. Why this particular value (0.5 eV) of potential difference across the nanoribbon was chosen?
5. What is the topology of the edge termination in the model (simple zigzag, bearded)? Can it be related to the experimental observations somehow?

Reviewer #2 (Remarks to the Author):

The manuscript entitled "Ballistic tracks in graphene nanoribbons" by J. Aproz et al. reports charge transport measurements on graphene nanoribbons grown on the sidewalls of SiC substrate steps. Hallmark of such ribbons, reported by the same group in 2014, include the possibility to observe conductance quantization in units of e^2/h , and ballistic transport over 16 μm distances. The explanation of the origin of these hallmarks are satisfying to some, and puzzling to others. In this work, authors dig deeper into the transport details along/across such nanoribbons and propose a feasible interpretation, via quantum transport simulations, for their measurements. The main text is well written and organized. Overall, I recommend this paper for publication in nature communications following minor revisions:

- 1) One element of novelty I find is the theoretical interpretation given in this manuscript. Another element has to do with growth that results in very smooth $(11-2n)$ facets. I encourage authors to elaborate on the growth conditions (e.g. Temperature, Pressure, Gases...) that lead to such reconstruction. Reproducing growth is the first step towards reproducing the exciting transport measurements on these ribbons (since 2014 it seems that such electrical measurements have not been reproduced outside the group of C. Tegenkamp). To what extent is growth related to smaller mean free path of $\sim 2\mu\text{m}$ compared to their seminal paper of 2014 (16 μm)?
- 2) Authors provide a conductance histogram for a "working" ribbon in Fig. 3d. It would be

illustrative to present a histogram for scans across several ribbons (like those shown in fig 2d). In other words, how frequently do you encounter the $R \sim 26 \text{ k}\Omega$ value when you scan over several areas of the chip?

3) For electrical measurements, the sharp tip has a radius of 40 nm (~middle of page 4). How certain are the authors that such probe allows them to probe the e^2/h ballistic track at the very bottom edge (there will be mechanical convolution when tip encounters the 23deg facet)? Even with a sharper Pt tip (~10nm radius) the later force on the tip (Fig. S3) is seen earlier than the height response when the tip encounters the (11-2n) facet.

4) Have authors performed similar measurements but on lithographically defined ribbons placed on a flat terrace? The two-probe setup is very elegant and deceptively simple, as it can carry many subtleties. In a very crude analogy to your setup, Y. Wu et al. ("Electrical transport across metal/two dimensional carbon junctions: edge versus side contacts" AIP Advances 2. 012132, 2012) show that gentle feedback of STM tungsten tip, used to contact graphene edges, can result in a variety of contact resistance values, $\sim 26 \text{ k}\Omega$ being one of them (fig.4 in that reference), depending on details of metal/graphene junction (e.g. tunneling or Sharvin resistance). It would be just comforting to know that e^2/h is not observed (or not as frequently and over long distances) on plain lithographic ribbons on flat terrace of SiC.

5) Ref. 23 has to updated.

6) Page 3, line 6: Please double-check notation for crystal planes. You first refer to crystallographically equivalent directions $\langle -1100 \rangle$. I am myself not sure if then planes should be denoted by $\{11-2n\}$ instead of (11-2n).

Reviewer #1 (Remarks to the Author):

The observation of quantized ballistic transport at ambient conditions and over large distances in one-dimensional structures formed in epitaxial graphene at the sidewalls of SiC substrate is highly intriguing. This work makes an important advance in investigating this phenomenon by means of spatially resolved two-probe measurements performed in a highly sophisticated experimental setup. In particular, this paper shows precise localization of different ballistic channels and their spatial segregation. In my opinion, the experimental part of this work merits publication in Nature Communications on its own. The interpretation of experiments is supported by a computational model that leaves more questions. Overall, the model shows agreement with experimental data, but I have a strong impression that any small change in model parameters may produce qualitatively different results. Furthermore, the choice of some of these parameters is hard to justify. I would encourage the authors to answer the questions below, provide arguments in support of the robustness of the results to small changes of input parameters, and possibly tone down the unambiguity of the theoretical interpretation of experiments. The questions are:

Answer: We thank the referee for carefully reading our manuscript and acknowledge his/her expert opinion. Indeed, the spatially resolved transport measurements are demanding and we appreciate the referee's recognition of this. The main concern of the referee is about the modeling of our experimental results. As a general remark, ab-initio calculations of such large system are not feasible, thus we were pleased to be able to catch most of the effects by a tight-binding modeling after considering the environmental effects. The parameters were checked carefully and chosen from experimental findings. However, since this was not stated clearly enough in the original manuscript, we have now made some clarifications to both the main text and supplementary material. Please see our point-by-point response below:

1. As Fig. 4a inset shows a portion of band structure with a precise location of the Fermi level. Is this choice of Fermi level somehow justified? How the results would change if the Fermi level is displaced by a certain amount (e.g. $0.05 t$)? It would be beneficial to the readers to present also the band structure plotted in a broader energy range and in the entire range of momenta.

Answer: Our results are very robust, once the Fermi energy lies within the range of a single (valence OR conduction) bulk sub-band. This was hinted in the initial submission, where we mentioned that higher order sub-bands tend to cluster states at only one side of the ribbon. However, we now state this explicitly in the main text, and demonstrate it in a new Supplementary figure (S4) and associated discussion. This figure also includes a "zoom out" of the band structure, as requested by the referee. This new figure shows that the three-plateau signature seen in experiment is robust over a range of energies near the Dirac point, and only fades once contributions from higher-order sub-bands become relevant. We note that the $0.05|t|$ displacement suggested by the referee (and also shown in this figure (light blue case)) is in fact quite large, and moves the system into the multiple sub-band regime.

Added to main text: "Furthermore this behaviour persists over a wide range of Fermi energies near the Dirac point (see further discussion in SI)."

2. The calculations seem to be performed at zero-bias conditions. Does this correspond to the experimental conditions?

The referee is correct -- our calculations are performed at zero-bias, unlike the experimental case where a finite bias is required. However, the magnitude of the bias does not play a role in the features discussed here, which arise due to changing the probe placement and not the bias voltage. This is clear from Fig 1(c) (inset) in the main manuscript, where constant slopes are seen for all voltages in the I-V curves, with different values emerging for different probe placements. This justifies the use of the linear response (zero-bias) limit in our calculations.

Added to main text: "The use of zero-bias linear response techniques is justified by the independence of the experimental conductance on the bias voltage magnitude, as evident from the inset in Fig 2c."

3. What is the physical mechanism behind the mass term?

Answer: The mass term in our simulations is an approximation to capture an effect previously observed by ab-initio simulations [Ref 30]. This is the suppression of states, other than the magnetic edge state, in a small energy window near the Fermi energy. This arises due to the hybridization of the edge of the graphene ribbon with the substrate, and was simulated in narrow ribbons where both edges connect to the substrate [Ref 30]. In our system, only the bottom edge has such a connection, so we limit the mass term to a small region (~ 10nm) near this edge. This is now clarified further in both the main text and Methods section.

Main text: (rephrased and added): In addition, ab initio studies of narrow nanoribbons with both edges bonded to the Si-face of SiC(0001) reveal that hybridization quenches states other than the magnetic edge state, leading to secondary gap opening near the Fermi energy. Only the lower edge in our system has such a bond, so we reproduce the effect by adding a sublattice-dependent gap-opening term only at sites near this edge (see Methods).

Methods: (added): This mimics the previously-noted effects of hybridization with the SiC(0001) surface.

4. Why this particular value (0.5 eV) of potential difference across the nanoribbon was chosen?

Answer: We know from scanning tunneling spectroscopy that the lower edge is neutral. On the other hand it is known from photoemission experiments that monolayer graphene on SiC is naturally n-type doped ($E_F - E_D \sim 0.5\text{eV}$) (e.g. Riedl et al [Ref 28] report 0.42 eV, but also slightly larger values are reported in literature). In our case, the upper edge of the GNR merges into the buffer layer, so it is reasonable to assume that the vicinity of the GNR to the buffer layer also suffers from unsaturated back bonds below the buffer layer. This will give rise to a non-uniform potential across the ribbon width, which we model as a linear drop of 0.5V across the ribbon. This point is also clarified further in the main text.

Added/clarified main text: To account, in a general way, for inhomogeneous potentials that arise due to the merging of the upper edge with the buffer region, and consequent charge transfer at this interface, we include a transverse electric field term which shifts the Fermi energy of the upper edge by approximately 0.5~eV relative to the lower edge. Monolayer graphene on SiC has been shown to be n-type doped by unsaturated bonds at the SiC interface. The merging of the top edge of the ribbon into the buffer layer should result in a similar local doping scenario.

5. What is the topology of the edge termination in the model (simple zigzag, bearded)? Can it be related to the experimental observations somehow?

Answer: We managed recently to measure with LT STM the lower edge. Due to the curvature, this is of course difficult and tricky. Nonetheless, the lower edge reveals nicely the zigzag structure. We provide here a STM image for the referee, but want to publish the STM data separately.

That the zigzag geometry used in the tight binding calculation is explicitly mentioned in the manuscript (first sentence, second paragraph on page 6).

Reviewer #2 (Remarks to the Author):

The manuscript entitled “Ballistic tracks in graphene nanoribbons” by J. Aproz et al. reports charge transport measurements on graphene nanoribbons grown on the sidewalls of SiC substrate steps. Hallmark of such ribbons, reported by the same group in 2014, include the possibility to observe conductance quantization in units of e^2/h , and ballistic transport over 16 μm distances. The explanation of the origin of these hallmarks are satisfying to some, and puzzling to others. In this work, authors dig deeper into the transport details along/across such nanoribbons and propose a feasible interpretation, via quantum transport simulations, for their measurements. The main text is well written and organized. Overall, I recommend this paper for publication in nature communications following minor revisions:

Answer: We thank the referee for his/her careful reading and acknowledge the very positive rating. Indeed, the modeling of such a large system within an inhomogeneous environment is computationally very demanding and we were happy to see that tight-binding can explain the main effects quite accurately. Detailed point-by-point answers are given in the following:

1) One element of novelty I find is the theoretical interpretation give in this manuscript. Another element has to do with growth that results in very smooth $(11-2n)$ facets. I encourage authors to elaborate on the growth conditions (e.g. Temperature, Pressure, Gases...) that lead to such reconstruction. Reproducing growth is the first step towards reproducing the exciting transport measurements on these ribbons (since 2014 it seems that such electrical measurements have not been reproduced outside the group of C. Tegenkamp). To what extent is growth related to smaller mean free path of $\sim 2\mu\text{m}$ compared to their seminal paper of 2014 (16 μm)?

Answer: We agree that the growth of the GNRs is delicate in a sense that easily other parasitic effects may occur, e.g. debunching of the sidewall facets. We also want to emphasize that both growth of ribbons and their electrical characterization were done independently in Walt de Heer’s group. This is also mentioned in our Nature 2014 paper.

Recently, we successfully transferred (and improved) the recipe to other laboratories for growing the ribbons, even on wafer scale. For the ribbons used in this publication, we used GNRs that were grown in collaboration with coauthor Alexei Zakharov from Lund. All experimental details are mentioned in Ref.23 (see your comment 5). The manuscript, headed by Alexei Zakharov, was submitted recently to Nano Letters. We hope that the referee is willing to wait for the other publication.. We want to emphasize that we have only made some minor changes to the standard recipe (details can be found in the revised version of our manuscript).

The mean free path of $\sim 2\mu\text{m}$ is related to residual steps of the SiC surface. This is mentioned, e.g. in Appl. Phys. Lett. 106, 043109 (2015), reference [3] in the manuscript.

2) Authors provide a conductance histogram for a “working” ribbon in Fig. 3d. It would be illustrative to present a histogram for scans across several ribbons (like those shown in fig 2d). In other words, how frequently do you encounter the $R\sim 26\text{k}\Omega$ value when you scan over several areas of the chip?

Answer: The deposition of Pt was done using a mask in an MBE system. The biggest problem is that we encountered with this technique was the unintended spreading of Pt, leading to a gradual transition of the electrode edge. This limited our investigation to regions where the spreading was not as bad (based on electrical measurements on the surroundings

of the ribbons) and for ribbons with larger mean free paths. The histogram shown in the manuscript is our best example. We have more measurements that show the bottom channel, but the lower grid resolution provides histograms dominated by the resistance of the surroundings. In addition, the ambient may affect the bulk channels, which are less robust compared to the edge state. We provide here one more example, where the resistance in the ribbon is slightly larger (35 k Ω). This might be due to a larger distance from the electrode than the mean free path (leading to a transmission factor $T < 1$). The scan is taken around 2-5 μm from the electrode.

Figure. (a) Resistance map of a graphene nanoribbon, extracted from I(V) curves similar to the one in (b). (b) I(V) curves recorded at the bottom edge (red), middle (blue), top (purple) and surroundings (black) of the ribbon. (c) Histogram of the resistance distribution in and around a graphene nanoribbon.

3) For electrical measurements, the sharp tip has a radius of 40 nm (~middle of page 4). How certain are the authors that such probe allows them to probe the e^2/h ballistic track at the very bottom edge (there will be mechanical convolution when tip encounters the 23deg facet)? Even with a sharper Pt tip (~10nm radius) the later force on the tip (Fig. S3) is seen earlier than the height response when the tip encounters the (11-2n) facet.

Answer: The referee should keep in mind that the GNR grows out of the surface. This model is supported by the STM image shown in context of Answer 5 to referee 1. Therefore, the curved tip, approached from the trenches of the SiC surface should provide a sharp conductance signal. In the AFM measurements, we have used tips with radius of curvature (R) < 5 nm (diamond tip) and ~10 nm (Pt tip). In both cases the set point force was kept to a minimum (1-2 nN), whilst the adhesion force calculated from Force vs Z-piezo displacement curves was found to be ~1.5 nN for the diamond tip and ~10 nN for the Pt tip. This gives a total force (F) of 2-3 nN for diamond and 10-15 nN for Pt. Based on the Hertz model $r = [3FR/4((1-\nu_s^2)/E_s + (1-\nu_t^2)/E_t)]^{1/3}$ [Surf. Sci. Rep. 1999, 34, 1-104 & Appl. Surf. Sci. 2007, 253, 3615-3626 & ACS Nano, 2015, 9 (3), pp 2843–2855], (where E_t , E_s and ν_t , ν_s are the elastic modulus and Poisson ratio of the substrate (graphene) and tip, respectively) the estimated contact area radius amounts to ~ 0.3 nm for the diamond tip and ~1 nm for the Pt tip. This small contact size (compared to the width of the ribbons) suggests that tip-induced convolution is kept to a minimum.

The supposed earlier increase of the lateral force signal compared to the height signal might be related to a slight overgrown ribbon.

4) Have authors performed similar measurements but on lithographically defined ribbons placed on a flat terrace? The two-probe setup is very elegant and deceptively simple, as it can carry many subtleties. In a very crude analogy to your setup, Y. Wu et al. (“Electrical transport across metal/two dimensional carbon junctions: edge versus side contacts” AIP Advances 2. 012132, 2012) show that gentle feedback of STM tungsten tip, used to contact graphene edges, can result in a variety of contact resistance values, ~ 26 kOhm being one of them (fig.4 in that reference), depending on details of metal/graphene junction (e.g. tunneling or Sharvin resistance). It would be just comforting to know that e^2/h is not observed (or not as frequently and over long distances) on plain lithographic ribbons on flat terrace of SiC.

Answer: We see the point of the referee and acknowledge his/her suggestion of further experiments. We do not have FIB etc. available in the system, so we cannot perform the envisaged experiments directly. In context of a different project, we prepared stripes of graphene on basis of epitaxial graphene on SiC(0001) with an external FIB. From all what we have seen so far we never measured an edge specific contact resistance of 26kOhm. For monolayer graphene systems, it is well known -also by many other investigations (e.g. F. Molitor et al. Semicond. Sci. Technol. 25, 034002 (2010)) - that lithography after growth of graphene generates rough edges giving rise to mobility gaps and Coulomb blockade effects. In our case, the edges – and in particular the lower edge – is smooth. This is supported by recent STM images, which will be published separately (see also answer 5 to the first referee).

5) Ref. 23 has to updated.

Answer: Please see above. The manuscript has been submitted. We provide the submitted manuscript for the referee.

6) Page 3, line 6: Please double-check notation for crystal planes. You first refer to crystallographically equivalent directions $\langle -1100 \rangle$. I am myself not sure if then planes should be denoted by $\{11-2n\}$ instead of $(11-2n)$.

Answer: We thank the referee for the careful reading. The convention is: $\langle \dots \rangle$ refers to a set of *directions*, while $[..]$ is a specific direction. Similarly, $(..)$ refers to a *specific* plane and $\{ .. \}$ refers to a *family* of planes. We refer to specific directions and changed the manuscript accordingly. Also, the notations in Figs.1 and 2 as well as Figs.S1 and S2 were corrected.

REVIEWERS' COMMENTS:

Reviewer #1 (Remarks to the Author):

The authors have clarified all the issues raised in my first report. I recommend for publication in Nature Communications the present revision of the manuscript.

Reviewer #2 (Remarks to the Author):

Having read the comments and improvements made by the authors, I recommend publication of this paper in nature communications

While I am not entirely satisfied by the response of the authors to my question # 3 ("...For electrical measurements, the sharp tip has a radius of 40 nm ...", I find sufficient elements of novelty and originality in this work. The response of authors might as well be valid, and this work should urge other groups to elucidate further details of epitaxial graphene nanoribbons. I hope publication of this work will continue to stimulate other groups to investigate these GNRs, which in my opinion remain puzzling and intriguing even after reading this work.

We thank both referees for their judgement and recommendation of acceptance of the revised manuscript.